# Usability Study of VR Interfaces for Learning from Demonstrations in Bimanual Tasks

Jeffrey Saber
*Vrije Universiteit Amsterdam*
Department of Computer Science
Amsterdam, the Netherlands
je_saber@live.concordia.ca

Muhan Hou
*Vrije Universiteit Amsterdam*
Department of Computer Science
Amsterdam, the Netherlands
m.hou@vu.nl

Kim Baraka
*Vrije Universiteit Amsterdam*
Department of Computer Science
Amsterdam, the Netherlands
k.baraka@vu.nl

*Abstract*—Learning from demonstrations (LfD) is a key component of state-of-the-art robot learning approaches that enable robots to learn complex tasks by observing and imitating human actions. While significant bodywork focuses on developing effective algorithms, demonstration quality remains a bottleneck in LfD, primarily due to suboptimal interfaces for collecting demonstrations. This paper addresses this gap specifically in the context of bimanual tasks by proposing a VR setup for demonstration data collection in which we compare two conditions: one in which the user teleoperates the robot with the robot always visible (teleoperation condition) and another in which the user demonstrates the task independently without visual feedback (egocentric condition). The task involves two Panda robot arms working collaboratively to pick up a tray stacked with cubes and place it at a designated goal. Performance is measured based on success rate and completion time. In addition, we conducted a user study to evaluate the user experience in VR environments. The data collected was then fed into a behavior cloning algorithm, where we analyzed training loss, validation performance, and error metrics such as Mean Squared Error (MSE) and Mean Absolute Error (MAE). Results suggest that the teleoperation system performs better in basic tasks, whereas the egocentric condition performs slightly better in complex tasks. The behavior cloning algorithm demonstrated that the teleoperation system had a more substantial generalization across all tasks compared to the egocentric system. The link to the code can be found here.

*Index Terms*—Learning from Demonstrations, Teleoperation Interfaces, Bimanual Tasks, Behavior cloning, Virtual Reality

## I. INTRODUCTION

There are many ways in which robots can be taught different tasks, such as reinforcement learning algorithms or deep learning. Learning from Demonstrations (LfD) became a revolutionary method for teaching robots new tasks due to its advantages. Research has shown that traditional mathematically based algorithms require expertise and an accurate real-world model [4]. LfD does not require expert knowledge in the domain, eliminating the complex mathematical models and computations needed for a traditional mathematical algorithm to work and simplifying the development process. Furthermore, LfD accelerates the learning process because it relies not solely on trial and error but uses demonstrations for faster convergence and more efficient learning [9]. It can be scaled to teach various tasks without redesigning the learning framework. This flexibility allows for applying LfD in diverse domains, from industrial automation to personal assistance.

In addition, it could include specific real-world examples that illustrate the practical benefits and effectiveness of LfD. Examples include robotic surgery, autonomous driving, and household robots performing daily chores. Finally, users and non-robotic experts can provide these demonstrations since they are intuitive—a natural human behavior.

Demonstrations can be provided in various ways, such as using a joystick, a keyboard, a virtual reality headset, or guiding a real-life robot through tasks. Using a simulation to provide demonstrations has many benefits, such as generating large amounts of data for machine learning algorithms at a low cost, accelerating the design process, and making it more efficient and safe for engineers and developers to use [6]. Numerous studies have shown that using a VR headset as input for demonstrations has been the most effective in terms of success rate and completion time [1] [3]. Moreover, VR headsets make it very convenient and user-friendly for users to provide demonstrations for the robot. The robot can clearly see the movements from different angles; no complex programming is needed; the robot just observes and imitates the real-world motions [1] [3].

Teleoperation teaching is the central Learning from Demonstration (LfD) method used in modern solutions [6]. Teleoperation is essential to robotics because human intervention should always be present. It is also used for tasks that require critical thinking and fast reflexes, for example, in a military mission [3] [10]. In this paper, a teleoperating system is developed in a VR simulation, using the VR controllers to guide the robots to complete a task. This system is compared with a baseline system in which the participants performed the task without having the robot visible.

This study compared two virtual reality environments developed using Pybullet with the Meta Quest 2, a virtual reality (VR) headset developed by Reality Labs. In the first environment, the user performs the task with the robot always visible, controlling it through the teleoperation system developed. In the second environment, the human performs the task independently; once the task is completed, the data and trajectory are recorded and fed to the robots afterward. In the teleoperation environment, the user can observe two panda robot arms, a tray, and some cubes. In contrast, the user can observe a tray and some cubes in the egocentric

environment. Users are asked to perform three tasks, each with increasing difficulty, to provide demonstrations for the robots. Once the demonstrations are completed, they will be fed into a behavior cloning algorithm, enabling the two robots to interact autonomously and perform bimanual tasks cooperatively.

This study aims to identify the most efficient approach to designing a VR environment that optimizes user experience and success rates. It also seeks to determine which method is the most user-friendly and effective in providing demonstrations within a simulation. Utilizing a simulation offers considerable flexibility in manipulating tasks and scenarios without additional costs. Several previous works have focused on developing systems for single-arm robot tasks and have been thoroughly discussed [1] [8]. While there is some understanding of how to build effective simulations, answering questions like whether the presence of the robot in the simulation is necessary or not would help develop new simulations for future work. The bimanual tasks provide a unique challenge due to the coordination required between multiple robotic arms, making them particularly suitable for assessing the efficacy of different LfD methods. By focusing on bimanual tasks, this study addresses a critical aspect of real-world robotic applications, where collaborative manipulation is often required for tasks such as assembly, manipulation of large objects, or surgical procedures. Specifically, the research question addressed is:

"How does seeing the robot perform the task within the VR simulation impact the user experience, success rates, time of completion, and performance of a behavior cloning algorithm of learning from demonstrations in bimanual tasks?"

The hypothesis states that the teleoperation system will yield a higher success rate and better results in the behavior cloning algorithm. However, regarding user experience and time to completion, the egocentric system is expected to perform better. This is because adapting to the robot's movements ensures that objects are always within reach; it also requires more time due to the unnatural nature of the movements. The contributions of this work can be summarized as follows:

1) We proposed a teleoperation system and compared it with a baseline.
2) We evaluated the user experience based on dizziness, intuitiveness, and general discomfort.
3) We uploaded the demonstrations in a behavior cloning algorithm to have a more detailed analysis of the environments' performances.

## II. RELATED WORK

### A. Interfaces for LfD on Bimanual Tasks

Several Frameworks and systems for bimanual tasks have been developed during the years, one example is the framework used to help robots for bimanual tasks in surgery [12]. Some systems implement Dynamic Movement Primitives (DMPs). DMPs are effective for adaptable robot trajectories in various tasks but struggle with new goal orientations, a

challenge partly addressed by leveraging invariance properties for fixed end-effector orientations [11]. In [11], the authors proposed the Target-Referred (TR) DMP implementation in which the learned trajectories are always expressed in a target reference frame, for instance, the frame attached to an object of interest, refer to paper [11] to learn more about this system. The tasks demonstrated in [11] were pick and place of large objects like a cardboard box, and turning a valve. These two papers utilize LfD to train robots for real-world bimanual tasks, rather than in simulated environments.

Other systems were developed for use in simulation, such as the OPEN TEACH system [5]. OPEN TEACH is a novel teleoperation system for bimanual tasks at low cost. In [5], the authors tested their system across various tasks, specifically 38 tasks, ranging from opening a drawer to making a sandwich. The authors wanted to do a variety of tasks, from using a single arm to using both arms, to test the diversity of their system. In this paper, the same equipment will be used, but the Pybullet library will be utilized, along with handlers to attach grippers to it, unlike the dexterity option proposed in the paper [5]. Furthermore, alternative systems use kinesthetic teaching [13], aiming to merge its advantages with teleoperation within a virtual environment.

Moreover, some systems, like Holo-Dex [15], a new framework for dexterous imitation learning, propose the use of mixed reality. In [15] the authors discuss how high-quality teleoperation can be achieved by immersing human teachers in mixed reality through inexpensive VR headsets. There are six different tasks discussed in this paper. Some examples are opening a bottle, sliding, and grabbing a card from the table. The tasks in [15] emphasize the use of dexterity, whereas this work concentrates on bimanual tasks.

Collectively, these papers introduce innovative approaches to teleoperation. In contrast, this study will conduct a detailed comparison of two specific environments, focusing on evaluating their performance and user preferences to provide a clearer understanding of their effectiveness and user satisfaction. Furthermore, this study focuses on bimanual tasks and having a good interface to teach two robot arms how to collaborate and cooperate to successfully perform complex tasks. LfD methods that utilize VR as a tool for collecting demonstrations have gained significant attention in recent research [17]–[21].

### B. Different Types of Inputs for VR

To provide demonstrations using VR, specific input methods are necessary. Examples include VR controllers, RGB cameras for full-body tracking, or specialized gloves for finger movement detection [2] [6] [7] [12]. In [7] the authors use a RGB camera to reconstruct the human pose in 3D, the visual feedback system allows the operator to make accurate judgments about the work site situation and thus easily control the robot to complete unstructured operational tasks [7]. Utilizing RGB cameras enables users to deliver precise demonstrations by employing full-body tracking, which accurately identifies the exact position of a human body in each time frame.

Another widely used VR input is the use of specialized gloves for finger tracking. As discussed in section 2.1, paper [15] uses gloves to track finger movements in addition to getting the position of the human hand. These gloves offer several advantages. Firstly, they allow users to move naturally without understanding how controllers work. Additionally, they are beneficial for robots with five fingers, such as the Pepper robot, enabling more intuitive and human-like interactions.

These papers use these different types of inputs because the robots used in their research require these special types of movements to be tracked, like the fingers on a hand, eye tracking or even full body movements. In this research, a panda arm with a hand capable of opening or closing will be used. To closely simulate the robot's hand, VR controllers will be used to attach URDF models, ensuring an accurate representation. Furthermore, using the original handlers will improve the tracking precision, hence optimizing the quality of demonstrations.

The quality of the demonstrations given to the robot is an important factor to consider while training a robot. A group of researchers have decided to study and teach humans how to give good demonstrations. They used different techniques in their study to allow users to give demonstrations [13], and their results show that having an expert do the task in front of the user and then having the user perform the task will lead to better and smoother demonstrations. Using their work, for this paper's experiment, an expert will initially perform the task while the user observes, subsequently, the user will perform the task to provide good demonstrations.

## III. METHODOLOGY

The teleoperation system and egocentric system were developed using two Python libraries: Pybullet, a physics simulation library, and Panda-GYM, which provides a collection of reinforcement learning environments. In Panda-GYM, the environment with the Franka Panda robot arm was used and edited to include two robots. These two libraries were selected for their popular use in physics simulations and flexible frameworks. To meet the requirements for this project, the two libraries were edited and adapted for bi-manual tasks.

### A. Teleoperation: System Architecture

The teleoperated system was designed with a flexible architecture to ensure scalability and ease of use. The system's goal is to help users navigate through the procedure of the tasks easily, efficiently, and most importantly, intuitively. In Figure 1 one can observe a simple flow chart of how the architecture is designed.

Controlling the Robot arms involves users moving their hands to the desired position. For this process to work, we had to extract some given data which are:

1) The human hand positions with respect to the original world frame. ($^{w}P_{hh}$)
2) The transformation Matrix from the new world frame to the original world frame. ($^{w}_{w'}T$)

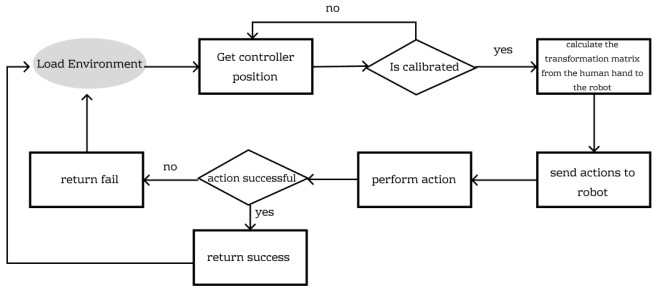

Fig. 1. Flow diagram of the Teleoperation architecture

3) The position of the human hand with respect to the new world frame is equal to the position of the robot end effector with respect to the old world frame (the goal that is needed). ($^{w'}P_{hh} = ^{w}P_{rh}$)

For context, the old world frame represents the origin (0,0,0) of the environment to which all displacements and actions are relative to. The new world frame has an origin at the human position. Getting the position of the human hand with respect to the new world frame the equation: $^{w'}P_{hh} = ^{w}_{w'}T^{-1} * ^{w}P_{hh}$ was used. After performing these calculations, we can feed the position of the new human hand position as an action to the robot arms. Additionally, an extra parameter is added to the actions, which involves closing and opening the gripper. Obtaining this extra parameter involves extracting the width of the URDF gripper model that was used for the human hand. The calibration process involves extracting the robots' end effector position and translating it to the new world frame. Two ghost cubes were added at the given positions, so users have to start from these positions.

For a visual representation, check out this video in which you can see the three tasks completed by one of the participants.

### B. Egocentric System

The egocentric environment provides users a more intuitive and seamless experience by allowing them to perform tasks naturally without needing to adjust their actions to accommodate the robots' movements. This design ensures that users can interact with the environment in a way that closely mimics real-world scenarios, thereby enhancing the overall user experience and the effectiveness of the simulation. In this setup, users effectively assume the position of the robots, eliminating the need for complex calculations or adjustments typically required when coordinating with robotic movements. By directly performing the role of the robots, users can engage with the tasks at hand as they would in a natural setting, leading to more authentic and fluid actions.

In Figure 2, a flow chart illustrates the architecture of the egocentric system, providing a visual representation of its workflow. The diagram outlines the process of the egocentric system.

After performing the task, in the first step, we evaluated if the task was successful or not. If the task was deemed

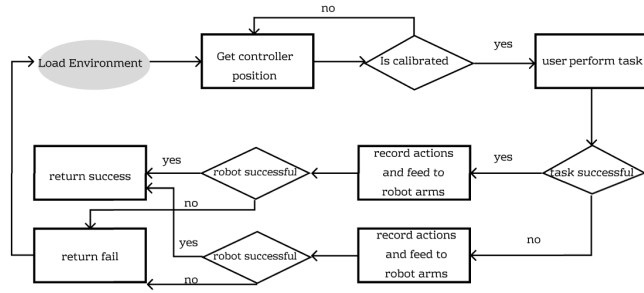

Fig. 2. Flow diagram of the Egocentric architecture

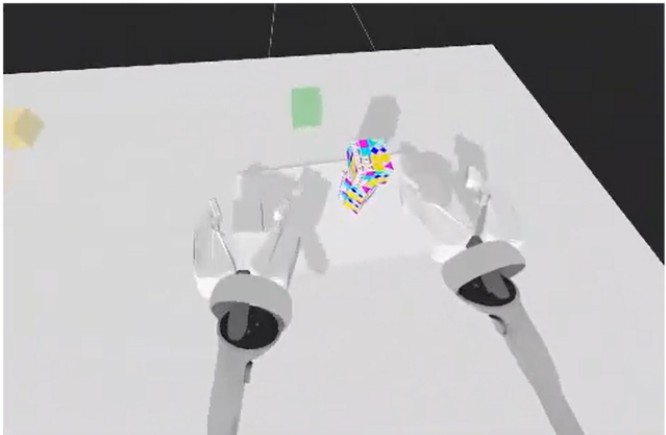

Fig. 3. Egocentric Environment

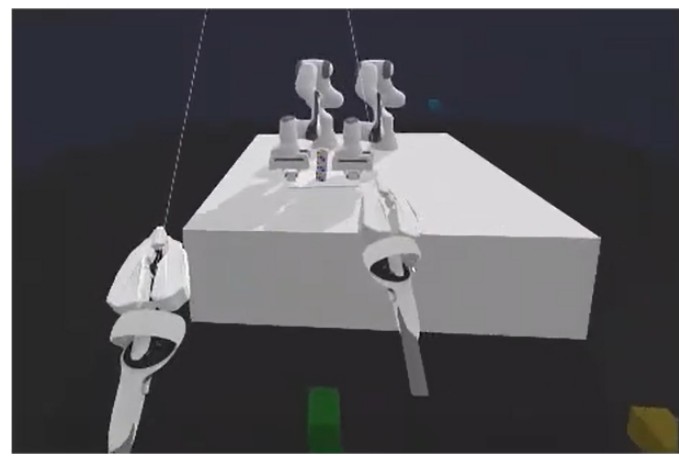

Fig. 4. Teleoperation Environment

successful, the user's actions were recorded and then fed back to the robots. Once the robots complete the actions, these actions would then be saved and stored for future use. On the other hand, if the user's trial was deemed unsuccessful, the actions were still fed to the robot. We then observed if the robot succeeded in completing the task because we were interested in the robot's actions.

For a visual representation, check out this video in which you can see the three tasks completed by one of the participants.

## IV. STUDY DESIGN

This study investigates the efficiency and user experience of two VR environments designed for bimanual tasks. The primary goal is to determine whether seeing the robot in action within the VR simulation improves user experience, success rates, and the performance of a behavior cloning algorithm. By comparing a teleoperation system with an egocentric approach, this research aims to identify the most effective method for providing demonstrations and optimizing the learning process in robotic applications. The following images depict the experimental environment. The first image represents the egocentric environment, while the second illustrates the teleoperation environment.

### A. Task Selection

Selecting a good task was a crucial part of this work. The task that will be chosen should meet five main conditions:

1) Bimanual Task: The task must require the use of both hands.
2) Real-World Use Case: The task should have practical, real-world applications.
3) Moderate Complexity: The task should balance being challenging enough to engage users without being overly complex.
4) Immediate Feedback: The task must provide instant feedback to the users.
5) Engagement: The task should be interesting and engaging for the users.

The task selected for this study was picking up a tray with objects on it, which meets all five criteria. Firstly, it is a bimanual task as it requires both hands to coordinate movements simultaneously to keep the tray stable during transport. Secondly, it has real-world applications, as many restaurants currently employ robots as servers. Thirdly, the task is moderately complex—it is more challenging than simple pick-and-place tasks but not overly difficult for users to complete. Fourthly, using a tray provides immediate feedback; users can observe and correct any tilting to improve their performance in subsequent trials. Lastly, the task is engaging and offers a small challenge, enhancing the overall user experience. After selecting the main task, three different conditions were added, to add a more general idea for the results. The first condition includes having only one cube, placing it on the tray, and picking the tray up to a specific goal. The second condition involves the user stacking three cubes on top of each other and picking the tray up without the stack falling apart. The third condition is similar to condition two but has an obstacle, making the user perform the task without dropping or touching the obstacle.

## B. Experimental Procedures

For this work, the experiments involved twenty human participants (sample size n = 20). All participants signed a waiver and agreed to comply with all regulations before participating in the study. They were then divided into two groups:

1) Teleoperation group: participants who used the teleoperation system and answered a survey based on their experience.
2) Egocentric group: participants who used the egocentric system and answered a survey based on their experience.

The participants were split randomly between the two groups, with ten participants for each group. At the start of the experiment, participants were warned that if they experienced any type of discomfort, they could stop the experiment at any time. Participants were then shown a video demonstrating how to perform the task, and they were given an explanation of what was expected of them. They are then given five minutes to play around the environment to get familiar with their surroundings. Once the experiment starts, the users will perform three tasks, having five trials for each task:

1) Task 1: Pick up one cube, place it on the tray, and place the tray at the designated goal.
2) Task 2: Stack three cubes on top of each other on the tray and pick up the tray to a designated goal.
3) Task 3: Stack three cubes on top of each other on the tray and pick up the tray to a designated goal while navigating around an obstacle.

For the teleoperation group, if the participant was able to complete the task successfully, the environment will reload automatically, collecting all the actions performed as well as the time of completion for each trial. Being able to monitor everything from the screen, if the cubes fall or the tray is not taken to the specific goal, we can manually reset the environment, collecting the same data but labeling it as unsuccessful. Similarly, for the egocentric system, the same procedure follows. However, after collecting the data, the script was run again, feeding all the actions performed by the participants to the robot arms. While observing the robot's arms' movements, a new set of data is collected.

After completing the three tasks, the users were asked to complete a survey about user experience and discomfort. The questions in the survey were specifically selected from a VR sickness questionnaire research [16]. In [14] the authors used the simulator sickness questionnaire (SSQ), which has been traditionally used for simulator motion sickness measurement, to measure the motion sickness in a VR environment. These questions are based on a Likert scale ranging from 1 to 3, with 1 being none and 3 being severe. Examples of the questions are "general discomfort," "dizziness with eyes closed," and "headache." Finally, all the data collected is discussed in Section 5.

## V. RESULTS AND DISCUSSION

During this study, the metrics used to evaluate the findings are success rates, time of completion, user experience, and

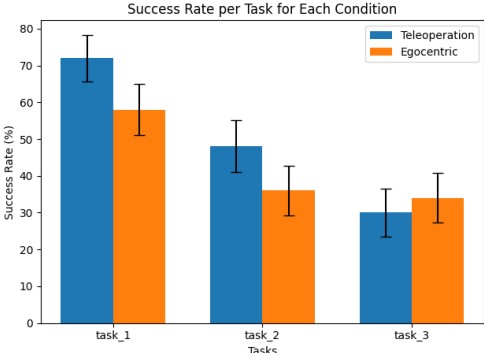

Fig. 5.   Success rates of both conditions for each task

finally, the performance of the data in a behavior cloning algorithm. These metrics were used to analyze the two environments and evaluate the better environment.

## A. Success rates and Time of completion

For every task, the success rates were measured. As seen in Figure 5, the success rates decrease per task, which is not surprising since the task increases in difficulty. The success rate for both environments was the highest for task 1, with teleoperation having 72% and egocentric having 58%. Surprisingly, in task three, the success rate for the egocentric system was higher. This suggests that when it comes to more challenging tasks, the egocentric system outperformed others. This is likely because users are accustomed to moving naturally, which helps them handle complex tasks more effectively.

For all three tasks, the p-values exceed the typical significance threshold of 0.05, indicating that there is no statistically significant evidence. In other words, the data show no significant effects or relationships for any of the tasks. To potentially find significant results, a larger sample size with more participants may be needed to increase the power of the tests.

The time of the successful attempts was measured in seconds to compare the task completion time. As seen in Figure 6, there is a big difference between the two environments in terms of the time of completion of the tasks, highlighting the difference in task 3. This shows that users in the egocentric environment had more intuitive controls and were more relaxed while completing their tasks.

The comparison of the time of completion as well as the success rate can show a noticeable pattern. Figure 6 shows that when users perform tasks without the robot visible, they will have a more natural response, hence completing the task quickly. Particularly in task three, for more complicated tasks, the participants almost took twice the amount of time. For the success rate, it was evident that having the robot visible would lead to a higher success rate, but surprisingly, for task three, this was not the case. As mentioned, this could result from users being more accustomed to their hands and having a more natural movement, allowing them to navigate obstacles.

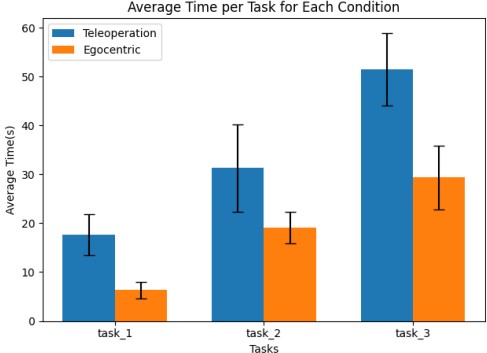

Fig. 6.  Average time taken for each graph for each task.

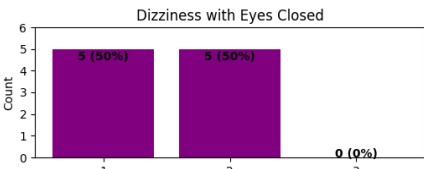

Fig. 7.  Results of the users' dizziness in the teleoperation architecture

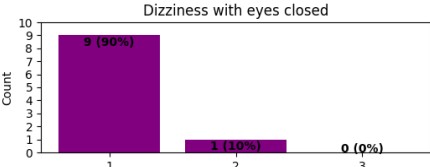

Fig. 8.  Results of the users' dizziness in the egocentric architecture

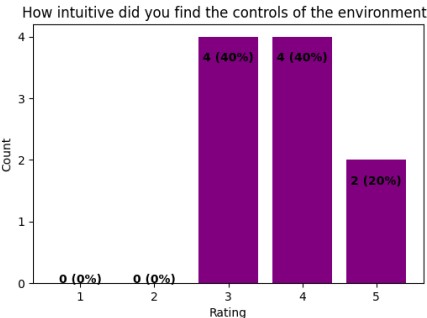

Fig. 9.  Results of the intuitiveness of the teleoperation architecture

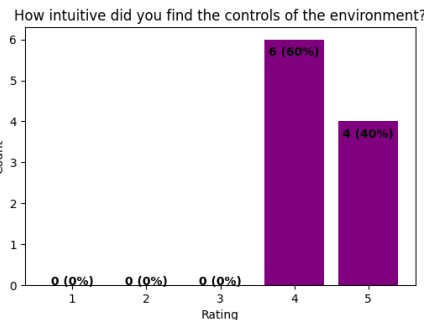

Fig. 10.  Results of the intuitiveness of the egocentric architecture

## B. User experience

After completing the three tasks, the users were then asked to answer a survey about their experience in VR and if they felt any discomfort. As observed in Figure 7, half of the users experienced dizziness in the teleoperation system with their eyes closed. This could be due to concentrating on the robot at all times and trying to understand its movement. On the contrary, focusing on your own hand will have you focused on one thing since, naturally, users understand how their hand works.

Another important factor worth noting is the intuitiveness of both environments. As shown in Figure 10, users in the egocentric environment found the controls more intuitive, leading to a better user experience. Having intuitive controls allows users to complete the task faster and more efficiently without having discomfort after completing the task.

After comparing the user experience, it is evident that users in the egocentric environment had a more pleasant user experience, allowing them to complete the tasks faster. This evaluation is shown through the figures stated above. The intuitive nature of controls within the egocentric environment reduces the cognitive load on users. By mimicking real-world movements and actions, the interface minimizes the learning curve typically associated with new technologies or complex interfaces. Users can seamlessly translate their physical movements into digital actions, enhancing their sense of control and streamlining the task completion process. A

more intuitive and familiar interface, such as the egocentric system, fosters a sense of comfort and confidence among users. This psychological comfort translates into improved task performance, as users feel more relaxed and capable of handling challenges presented by the tasks.

## C. Behavior Cloning Algorithm

Behavior cloning is a supervised learning approach where an agent learns to replicate expert actions. It does this by mimicking expert behavior through a dataset of expert demonstrations. The results of a behavior cloning algorithm provide a detailed understanding of the decision-making process, offering insights into the underlying policy and behavioral patterns. Unlike a success rate analysis, which merely indicates the proportion of successful outcomes, behavior cloning reveals the strategies and decision context, allowing for deeper analysis and continuous improvement. This method enables generalization to unseen states and facilitates error analysis by pinpointing where deviations from human actions occur, thus providing a richer and more comprehensive framework for interpreting and refining policies.

Key hyperparameters include 64 LSTM units per layer, which capture temporal dependencies. The dropout rate of 0.2 prevents overfitting by randomly dropping input units during training. The learning rate of 0.001 controls the optimizer's convergence speed. The batch size of 32 affects training stability and speed.

The model is trained over 50 epochs. A validation split of 0.3 is used to monitor performance and prevent overfitting. The

| Metric | Task 1 | Task 2 | Task 3 |
|--------|--------|--------|--------|
| Train MSE | 0.0578 | 0.0560 | 0.0516 |
| Train MAE | 0.0997 | 0.0979 | 0.0990 |
| Validation MSE | 0.0532 | 0.0398 | 0.0351 |
| Validation MAE | 0.0340 | 0.0255 | 0.0284 |

TABLE I
METRICS FOR DIFFERENT TASKS IN TELEOPERATION

| Metric | Task 1 | Task 2 | Task 3 |
|--------|--------|--------|--------|
| Train MSE | 0.0673 | 0.0742 | 0.0815 |
| Train MAE | 0.0931 | 0.0993 | 0.1116 |
| Validation MSE | 0.0539 | 0.0590 | 0.0119 |
| Validation MAE | 0.0206 | 0.0243 | 0.0285 |

TABLE II
METRICS FOR DIFFERENT TASKS IN EGOCENTRIC

process begins with collecting the demonstration data. This data is normalized using 'StandardScaler' and reshaped to fit the LSTM input requirements. The model comprises LSTM layers for temporal dependencies, dropout layers for regularization, and a dense output layer. The model is compiled with Mean Squared Error (MSE) as the loss function, and the Adam optimizer is used for weight updates. Training involves fitting the model to the data, and performance metrics like MSE and Mean Absolute Error (MAE) are tracked.

The tables below show the MSE and MAE values for the validation and training. The results indicate that the teleoperation system generally has lower training and validation errors compared to the egocentric system across all tasks, suggesting better performance in the teleoperation system.

Figures 11 and 12 show the training and validation loss curves for each task in the teleoperation system. Both graphs indicate a strong initial fall, indicating that the model quickly learns the fundamental patterns in the training data. Task 1 stands out for its rapid convergence, which could be attributed to simpler patterns or more accurate representation in its training data than other tasks.

Figures 13 and 14 depict the training and validation loss curves for tasks in the egocentric system, revealing interesting insights into how the model learns. Both curves show a sharp initial decline, suggesting the model rapidly learns the underlying patterns in the training data. Task 3 particularly stands out for its fast convergence, which could be due to either simpler patterns or more accurate representation in its training data compared to other tasks.

## VI. CONCLUSION

Based on our study comparing two VR environments for providing demonstrations, our findings highlight significant insights. The egocentric system emerged as notably user-friendly, offering a straightforward and intuitive experience. However, the teleoperation system demonstrated superior performance metrics, including a higher success rate and en-

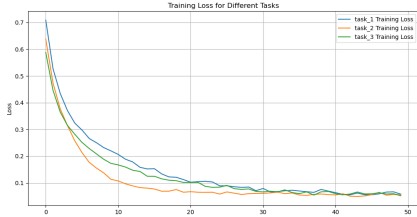

Fig. 11. Training loss in teleoperation architecture.

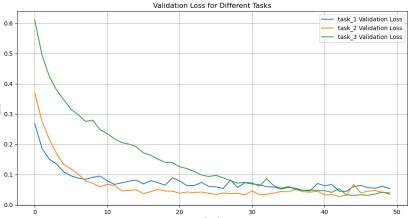

Fig. 12. Validation loss in teleoperation architecture.

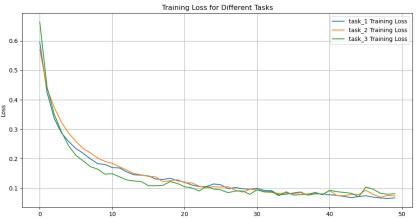

Fig. 13. The training loss in egocentric architecture.

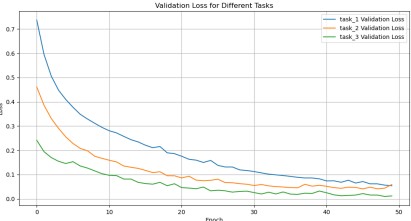

Fig. 14. The validation loss in egocentric architecture.

hanced generalization within behavior cloning algorithms. These results underscore the teleoperation system's potential as an ideal choice for scenarios requiring robust and adaptable demonstration capabilities in virtual reality settings. With more fine-tuning and a more accurate egocentric environment, it would be possible to have an egocentric environment that performs as well as a teleoperation system, increasing the user experience and time efficiency. More research and development is needed. Future work could explore integrating haptic feedback to enhance user immersion and interaction within both VR environments. Additionally, another study could be conducted with an autonomous arm, and the users would try to cooperate with that arm as a third condition. This would be particularly useful if a company bought a new arm; then, the company would not have to reteach the task for both arms, but the new arm would learn how to cooperate with the old

arm. Finally, more participants should be gathered and divided into several groups, like those who have experience with VR or experience with the panda arm, to see if this would affect the results.

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
