# OpenReview forum: "Usability Study of VR Interfaces for Learning from Demonstrations in Bimanual Tasks"
_humanrobotinteraction.org/HRI/2025/Workshop/VAM — HRI 2025 Workshop VAM Submission_

### Official Review · Reviewer_xe5m · 2025-02-28

**Rating:** 7
**Confidence:** 5

**Review:**

This paper details the development and testing of two VR systems to find which is better for training robots in a two-handed task using the ‘learning from demonstration (LfD)’ method – i.e. feeding data of humans performing a motion to a robot for them to replicate. These two VR systems are, one, a teleoperation system where the user remotely controls the robotic arms to lift a tray, and two, an egocentric system where the user interacts with the tray directly. The authors find that while the teleoperation system produced more accurate and easily translatable motions, users (N = 20) were faster in and generally preferred the egocentric system.



The contributions towards VR systems for LfD are quite interesting especially in regard to usability for human users vs. results for the robotic arms. It would be interesting to see where this research leads and if there is an optimal sweet spot between the teleoperation and egocentric systems that satisfies all metrics of success. Or if the teleoperation usability score improves over a prolonged period of use (i.e. as the users get used to it).



I quite like this paper, but think it could do with a good proofread. There are several areas where words have been skipped on accident (i.e. in the abstract, “While there is a large body work…” should be “While there is a large body of work…”). There is also inconsistent use of acronyms, with ‘learning from demonstration’ sometimes shortened to LfD or LFD. Also, once an acronym is introduced, it is jarring to go back to the full term. In the grand scheme of things, these grammatical errors do not take away from the content of the paper, but do make it a little difficult to read.

---

### Decision · Program_Chairs · 2025-02-26

Accept